# Diffractive Refractometer Based on Scalar Theory

**DOI:** 10.3390/polym15071605

**Published:** 2023-03-23

**Authors:** Marcella Salvatore, Francesco Reda, Fabio Borbone, I Komang Januariyasa, Pasqualino Maddalena, Stefano Luigi Oscurato

**Affiliations:** 1Centro Servizi Metrologici e Tecnologici Avanzati (CeSMA), University of Naples “Federico II”, Complesso Universitario di Monte Sant’Angelo, Via Cintia 21, 80126 Naples, Italy; 2Physics Department “E. Pancini”, University of Naples “Federico II”, Complesso Universitario di Monte Sant’Angelo, Via Cinthia 21, 80126 Naples, Italy; 3Department of Chemical Sciences, University of Naples “Federico II”, Complesso Universitario di Monte Sant’Angelo, Via Cinthia 21, 80126 Naples, Italy

**Keywords:** refractive index, diffraction gratings, scalar diffraction theory, azopolymers, soft lithography

## Abstract

The measurement of the refractive index typically requires the use of optical ellipsometry which, although potentially very accurate, is extremely sensitive to the structural properties of the sample and its theoretical modeling, and typically requires specialized expertise to obtain reliable output data. Here, we propose a simple diffractive method for the measurement of the refractive index of homogenous solid thin films, which requires only the structuring of the surface of the material to be measured with the profile of a diffraction grating. The refractive index of an exemplary soft-moldable material is successfully estimated over a wide wavelength range by simply incorporating the measured topography and diffraction efficiency of the grating into a convenient scalar theory-based diffraction model. Without the need for specialized expertise and equipment, the method can serve as a simple and widely accessible optical characterization of materials useful in material science and photonics applications.

## 1. Introduction

The refractive index *N* is a complex number that describes the optical properties of materials [1]. Microscopically, the refractive index characterizes the response of the polarizing atoms and molecules composing the material to the electromagnetic field of incident radiation. Macroscopically, the refractive index ultimately describes the effect of a material on an interacting light field. From ray optics to the electromagnetic treatment of light, the real part *n* and the imaginary part *κ* of the refractive index determine the ability of material interfaces to bend light rays, the velocity of the light in the bulk medium, the attenuation of propagating light due to material absorption, and the reflection and transmission characteristics of interfaces between different media, to name a few factors they determine [1,2]. In wave optics, the spatial modulation of the complex refractive index of a material on the scale of the wavelength of light can also be seen as the ultimate origin of the phenomenon of light diffraction, causing spatially selective absorption and wavefront retardation [1]. The measurement of the refractive index *N = n* + *i κ* and its wavelength dependence, called dispersion, in solid or liquid materials is of fundamental relevance for applications in many fields, including bulk [1] and emerging flat and metasurface optics [3,4,5], biological and biomedical research [6], and the chemical analysis of solutes in liquids [7].

The state-of-the-art measurement technique for solid material samples, especially in the form of thin films, is optical ellipsometry [8], which determines the optical constants (i.e., the refractive index *n* and the extinction coefficient *κ*) in a broad wavelength range based on the attenuation and change in polarization of light that occurs when light is reflected at the surface of a material sample. To be effective, ellipsometry requires the construction of an optical model for the sample structure from the measured optical data, the suitability of which is critical to avoid artifacts or large errors in the final results, especially when high accuracy is desired. One of the drawbacks of ellipsometry is its high sensitivity to the sample geometry, including the surface roughness and volume homogeneity, which requires advanced modeling techniques and ultimately independent sample characterization to maintain high data reliability, with a critical role also played by the experienced user [9]. In addition to ellipsometry, other optical techniques can measure the optical constants of solid and liquid materials. Standard absorption spectroscopy using dedicated spectrophotometers can provide measurements of the extinction coefficient *κ* in a wide spectral range with high accuracy, while the refractive index *n* can finally be measured via geometrical [10], interferometric [6,11], or reflectance methods [12,13], which are particularly suitable for the analysis of liquid [14,15] and scattering samples. 

Diffractive refractometers based on diffraction gratings or photonic crystal fibers have been proposed to measure liquid materials surrounding the diffractive optical element [16,17,18], but light diffraction has rarely been used to build refractometers for characterizing solid materials. Despite the clear dependence of the efficiency of a diffractive device on its refractive index *n*, the parametric form for this dependence made explicit in simple theoretical models is rarely achievable under experimental conditions. In the general case, the model that quantitatively describes the diffraction efficiency can instead be very complex and computationally demanding, with additional complications imposed by device fabrication defects that can make the analysis for refractometry untenable. Scalar diffraction theory, although approximate in some aspects when compared to the full electromagnetic theory of diffraction, can provide a simple parametric model to determine the refractive index *n* of the solid dielectric material used to fabricate a diffractive element from the measure of its diffraction efficiency and the characterization of its surface profile.

Here, we present a general framework for broadband refractive index measurement based on the analysis of the diffraction performance at different wavelengths of light of a transmissive sinusoidal diffraction grating, taken as an example of a possible device with structural properties falling within the scope of scalar diffraction theory. The method is potentially applicable to any unknown homogeneous dielectric material compatible with a simple replica molding process capable of replicating the surface pattern of a lithographically fabricated diffraction grating. A sinusoidal surface relief Grating (SRG) fabricated on the surface of an azopolymer thin film by two-beam interference lithography is used here as a controllable and cost-effective master for grating fabrication.

## 2. Materials and Methods

### 2.1. Grating Fabrication

Light-induced SRGs, used as master for the fabrication of PMMA gratings, were inscribed on the surface of an azopolymer thin film using two interfering p-polarized laser beams at *λ* = 491 nm with an average incident intensity of about 185 mW/cm^2^ over a spot diameter of ~4.0 mm. The PMMA (Merck KGaA, Darmstadt, Germany, Mw 10000, 4% wt. solution in 1,1,2,2-tetrachloroethane) grating was fabricated using a two-step replica molding technique [19,20,21] in which a PDMS (Sylgard 184—Dow Corning, Midland, Michigan, USA precursor: curing agent = 10:1 *w*/*w*) mold reproducing the complementary surface texture of the master was first fabricated and then used for PMMA patterning in the second step. The amplitude of the replicated grating was controlled by adjusting the exposure time of the azopolymer master during the SRG inscription.

### 2.2. Diffraction Efficiency Measurement

The relative diffraction efficiencies of the grating were calculated by measuring the laser power in the propagating diffraction orders using a NOVA II power meter (OPHIR photonics, Darmstadt, Germany) equipped with a PD300-3W photodiode sensor (sensitivity spectral range 350–1100 nm). To demonstrate the working principle of the method with a monochromatic light source, a TE-polarized He-Ne laser at *λ* = 633 nm was used to irradiate the grating from the substrate side at normal incidence. The broadband measurements were instead collected at several discrete wavelengths (450–900 nm) using a supercontinuum TE-polarized laser source (SuperK COMPACT—NKT Photonics, Birkeroed, Denmark) in the same incidence configuration. A set of FKB—Thorlabs bandpass filters (FWHM 10 nm) was used to select the narrow wavelength band centered on the nominal wavelength of interest. The spot size was maintained at ~2 mm in diameter by means of an adjustable telescope collimating the beam at different wavelengths and a fixed iris placed in front of the sample. 

### 2.3. AFM Characterization

The surface topography was characterized using an Atomic Force Microscope (Alpha RS 300, WITec, Ulm, Germany) operated in tapping mode. High-aspect-ratio probes (ISC-225C3_0-R, Team Nanotec, Villingen-Schwenningen, Germany) mounted on a cantilever with a resonance frequency of 75 kHz and a spring constant of 3 N/m were used to reduce errors in the quantitative topographic characterization. Typical AFM scans for grating profile measurements were taken over a sample area of 40 × 0.8 μm^2^ with a resolution of 2000 × 40 px^2^.

## 3. Results

### 3.1. Refractive Index in the Scalar Diffraction Theory of Phase Gratings

To illustrate the working principle of the proposed diffractive refractometers, we first review some general aspects of the diffraction behavior of standard surface diffraction gratings, which are built as periodic reliefs on the surface of an isotropic dielectric material, as schematized in Figure 1 for the case of a sinusoidal grating.

A surface diffraction grating splits the incident radiation into different beams (diffraction orders) whose propagation direction and relative intensity depend on the incident light wavelength *λ*, the refractive indices of the grating *n* and the surrounding material *n_s_*, and the details of the grating geometry, including the actual surface profile *s*(*x*), the periodicity Λ, and the modulation amplitude *h* of the periodic grating grooves (Figure 1a). The grating equation [1] defines the angle *θ_m_* of the propagating diffraction orders for an incident plane wave with wavevector $k_{i}=ksin\;\theta_{i}$ as $k_{m}=k_{i}+mG$, where k=2π/λ; G=2π/Λ and m=0, ±1, ±2,⋯. For normal incidence ($\theta_{i}=0$), the condition $\left|m\right|<\Lambda /\lambda$ defines the maximum number M of propagating orders allowed. According to this relation, the diffraction angles $\theta_{m}=\mathrm{asin}\left(mG/k\right)$ for normal incidence depend on the groove periodicity Λ, while they are independent of the details of the grating structure, including the refractive index *n* of the grating material, the surface profiles, and other structural parameters (e.g., the modulation amplitude, the thickness of the unmodulated material layer, the possible presence of a substrate, etc.). Instead, the structural and chemical properties affect the diffraction efficiency of the grating *η_m_*, which is a measure of the fraction of the incident light power that is converted into a specific propagating diffraction order.

A comprehensive quantitative understanding of the diffraction efficiency in the general case, involving absorbing, inhomogeneous and anisotropic materials, arbitrary periodic surface profiles, and periodicity-related and polarization-dependent effects, requires a full electromagnetic treatment of the light-matter interaction in the grating [22]. However, in many cases the diffraction behavior of a surface relief grating made of a homogeneous dielectric material can be successfully described within the framework of the scalar diffraction theory [23,24]. In addition to requiring a significant reduction in modeling and computational resources, the scalar diffraction theory preserves a simple parametric dependence of the diffraction efficiency on the grating characteristics, including the surface profile and the refractive index of the grating material. This theory potentially allows the retrieval of information regarding the morphology and the refractive index of the grating to be extracted from the quantitative measure of its diffraction efficiency. However, there are several theoretical and experimental aspects to be considered for the direct application of the scalar model for a refractometer. These include the operation of the grating within the range of the validity of the scalar model and the possibility of incorporating the details of the actual grating profile, as they result from the characteristics of the lithographic process used for fabrication and possible fabrication defects. The scalar theory is generally accepted to provide a sufficiently accurate diffraction efficiency, compared to fully vectorial electromagnetic theories, only for relatively simple surface profiles (e.g., sinusoidal, rectangular, triangular), at quasi-normal incidence, and for low λ/Λ and low h/Λ ratios [25]. Among the simple geometries falling within such limits, sinusoidal surface grating profiles are of particular practical interest. First, the analytic diffraction behavior of these gratings is completely described by the smallest number of geometrical parameters, requiring only the specification of the periodicity Λ and the amplitude h of the sinusoidal surface profile $h\left(x\right)=0.5\left(1+\sin\left(2\pi x/\Lambda \right)\right)h$. Second, other periodic surface profiles can be described as a superposition of sinusoids via Fourier decomposition, allowing a simple extension of the analytical results of ideal sinusoidal gratings to more complex surface geometries, possibly including deformations or manufacturing defects [26]. In addition, (quasi-ideal) sinusoidal surface profiles in the range of applicability of the scalar diffraction theory can be readily fabricated using interference lithography, making them easily and cheaply available for applications [27].

Figure 1a shows the schematic of the diffraction behavior of an ideal sinusoidal surface grating. In the simplest geometry suitable for scalar diffraction theory, the sinusoidal grating is assumed to be a boundary surface between two semi-infinite half-spaces, consisting of a dielectric material with refractive index *n* and a surrounding material with refractive index *n_s_* (air is considered here, so $n_{s}=1$). For a plane wave of amplitude *A_0_* normally incident on the grating (Figure 1a), the diffraction efficiency $\eta_{m}={\left|A_{m}\right|}^{2}/{\left|A_{0}\right|}^{2}$ of the *m*th diffraction order is expressed analytically in terms of Bessel functions of first kind of order *m*, as:(1)$$\eta_{m}={\left|J_{m}\left(\beta /2\right)\right|}^{2}$$

According to Equation (1), the parameter
(2)$$\beta =\pi \left(n-1\right)h/\lambda$$
completely defines the diffraction efficiency of each order. This can be interpreted as the maximum phase delay accumulated in the surrounding medium by the incident plane wavefront as it propagates through the grating structure. Figure 1b shows the behavior of Equation (1) as a function of β for the first five diffraction orders. 

From Equations (1) and (2), it is easy to see that, for a given grating amplitude *h* and incident wavelength *λ*, the diffraction efficiency in any order *m* depends only on the refractive index *n* of the grating material. This simple consideration then gives a potential strategy to directly measure the real part of the refractive index *n* of the grating material by (numerically) reversing Equation (1) through an accurate measurement of the grating depth *h* and the diffraction efficiency in one of the propagating orders. To achieve higher accuracy in the estimation of *n* at the specific light wavelength *λ*, the grating depth could eventually be optimized to operate in a range of maximum sensitivity $S_{m}=\left|\partial \eta_{m}/\partial \beta \right|$ for efficiency variations in the considered order with respect to the parameter β (Figure 1c). Maximum sensitivity is expected in the 0th for *β* = 2.16. However, the typical experimental conditions, as schematized in Figure 1d, prevent the direct use of Equations (1) and (2) in this simple form for a reliable estimation of *n*. First sources of discrepancy are the portions of light power reflected at each interface of a real device, which are not accounted for in the simple scalar picture of Figure 1a. In addition to the simple effect on the absolute value of the diffraction efficiency, the reflectance of the devices depends on the refractive index of the grating material itself, which, although it typically gives a negligible practical contribution due to the small power carried by the reflected fields, makes the unambiguous inversion of Equation (1) impossible. In addition, a physical grating has a finite thickness, possibly non-zero absorption at the probe wavelength, and requires the presence of a supporting substrate, the presence of which further alters the transmitted and reflected light power. The inclusion of all these effects makes the use of ellipsometry a typical necessity for the characterization of the optical constants of solid material layers in the general case. In addition, the surface profile of a real diffraction grating is never perfectly sinusoidal due to unavoidable fabrication errors common to any lithographic method used for its fabrication, which further deviate the absolute diffraction efficiencies $\eta_{m}^{\prime }={\left|A_{m}^{\prime }\right|}^{2}/{\left|A_{0}\right|}^{2}$ in experiments from the predictions of Equation (1). 

Fortunately, the scalar diffraction theory, which differs from fully vectorial methods, offers simple ways to overcome these limitations, making a direct refractive index measurement from diffraction analysis effectively feasible. First, the real surface profile *h*(*x*) of the grating, as measured for example by using an Atomic Force Microscope (AFM), can be retained in the calculation of the diffraction efficiency [26,28]. Second, the effect of the finite grating thickness and the reflections at each interface in the light path can be averaged out by simply defining the (relative) transmitted diffraction efficiency as the ratio of the intensity in propagating in the diffraction order *m* and the intensity $A_{0}^{*}$, measured in all the propagating diffraction orders as $\eta_{m}^{*}={\left|A_{m}\right|}^{2}/\sum_{m}^{M}{\left|A_{m}\right|}^{2}={\left|A_{m}\right|}^{2}/{\left|A_{0}^{*}\right|}^{2}$, as schematized in Figure 1e. This alternative definition of the diffraction efficiency also removes the sensitivity of the method to the absorbance of the material (and hence to the extinction coefficient *κ*) at the operating wavelength, and it is compatible with the conservation of the irradiance in the interaction of the incident light wave with the diffractive surface within the scalar diffraction theory, which is also valid in the non-paraxial approximation [22,25,29,30].

We have recently shown that this simple generalization of the scalar theory can be successfully applied to the quantitative description of the diffraction behavior of real (quasi-sinusoidal) Surface Relief Gratings (SRGs), inscribed directly on the surface azobenzene-containing materials [31,32,33] by interference lithography [28,34,35,36,37,38,39,40] and Digital Holography [41,42,43]. However, the results of this scalar model are, however, applicable to any diffraction grating made of a dielectric material, provided that the scalar approximation remains valid. Within such a model, the relative diffraction efficiency for the *m*th propagating diffraction order is explicitly related to the actual surface profile *h*(*x*) of the grating by a simple relation [22,26,28]: (3)$$\eta_{m}^{*}={\left|\frac{1}{\Lambda }\int_{-\Lambda /2}^{\Lambda /2}e^{i\frac{2\pi }{\lambda }\left(n-1\right)h\left(x\right)}e^{-im\frac{2\pi }{\Lambda }x}dx\right|}^{2}\;$$

Similar to the case of ideal sinusoidal gratings, $\eta_{m}^{*}$ in Equation (3) retains the explicit parametric dependence on the refractive index *n* of the grating, even though *h*(*x*) may eventually differ from a sinusoid. A fitting technique can then be used to extract the best estimate for the unknown parameter *n* from the accurate measure of the relative diffraction efficiency produced by the grating and the results of the scalar model in Equation (3).

### 3.2. Workflow of the Method

The workflow of the proposed diffraction-based information retrieval method for the measurement of *n* at the wavelength *λ* is shown in Figure 2. In the first step, the surface of the material to be measured is structured with a 1D (quasi-sinusoidal) diffraction grating, having a geometry compatible with the scalar diffraction approximation. Then, the grating is characterized by the measure of the relative transmitted diffraction efficiency in one of the propagating orders and by the topographic analysis of the surface profile, both of which enter into the fitting process of *n* using the scalar diffraction model of Equation (3). In principle, a single high-resolution surface profile measured on a large homogeneous grating and a single measure of the relative diffraction order may be sufficient to achieve a good fit convergence. In this case, the accuracy of the refractive index estimation is mainly determined by the grating depth *h*, with a sensitivity behavior similar to that of the ideal sinusoidal grating described in Figure 1c. However, possible inhomogeneities in the profile depth and/or shape of the grating, e.g., introduced during fabrication, may impose a trade-off between the optimal sensitivity to the diffraction efficiency variations and the number of independent estimates of the method necessary to average the inhomogeneities. As discussed below, grating inhomogeneities are the ultimate source of uncertainty for the refractive index measurement using this method.

### 3.3. Measurement of the Refractive Index of PMMA

To demonstrate the validity of the proposed method, we measure the refractive index of a poly(methyl methacrylate) (PMMA) grating at different discrete wavelengths in the range 450–900 nm, using the 0th diffraction order for maximum accuracy (Figure 1c) in the scalar model fit. We selected this material for a reliable comparison of our results with a large literature of refractive index data measured via ellipsometry [44,45]. In addition, PMMA is also considered as a prototype of dielectric moldable material to which the proposed method can be applied, being fully and directly compatible with soft lithographic replication of the quasi-sinusoidal SRGs fabricated on azopolymer thin films via interference lithography, used here as a master for grating molding [31]. The description of the light-induced fabrication of the azopolymer SRG for the master quasi-sinusoidal grating and its transfer onto the PMMA layer is given in Experimental Methods, while additional details on the polymer for the surface structuring process and its applications can be found in our previous works [41,46,47,48,49].

Figure 3a shows the AFM micrograph of the quasi-sinusoidal PMMA grating resulting from the replica molding process. The quasi-sinusoidal surface profile has a periodicity Λ ≈ 2.04 µm and a modulation depth of approximately 480 nm. Such a grating amplitude results in a good compromise between an optimized sensitivity of the diffraction efficiency to the refractive index variations in the whole wavelength range (*β*~1.7–0.84) and the necessity to be in a monotonically decreasing range for the zeroth order diffraction order (Figure 1b). The latter allows the accurate identification of the region of the grating for both optical and topographic analysis, which has an inhomogeneous gaussian depth due to gaussian beam profiles of the interfering light beams used to fabricate the azopolymer SRG master. The target grating region for the analysis is then uniquely identified via the simultaneous minimum zeroth order light diffraction efficiency and the maximum surface modulation depth in the grating topography. However, to fully characterize the consistency of the method, *J* = nine different AFM profiles $h_{j}\left(x\right)$ around the region of the topographic maximum over an area approximately 2 mm in diameter (comparable to the size of the probing diffraction beam) were collected and used for the independent diffraction efficiency calculation in the model of Equation (3). To further reduce errors due to the discretization of the measured AFM profile, a third-order Fourier series fit of the quasi-sinusoidal grating profiles is used in the effective diffraction calculation, as described in detail elsewhere [28].

To illustrate the working principle of our method, we first measure the diffraction efficiency of the grating for a He-Ne laser of a wavelength of 633 nm (Figure 3). To increase consistency, five independent measurements of the relative diffraction efficiencies $\eta_{m}^{exp}$ in the first seven orders were individually averaged by repositioning the grating in the target area each time. The relative diffraction efficiency measured in the first five diffraction orders is shown in Figure 3b, while the power in the *m* = ±3 orders was below the detection sensitivity of the optical power meter and was neglected in the analysis. As expected from the quasi-sinusoidal surface profile, asymmetric diffraction efficiency is observed in the homologous (*m* = ±1 and *m* = ±2) diffraction orders [28], which further strengthens the necessity to use the extended scalar model in Equation (3) for the accurate analysis of the diffraction of the considered grating.

According to the workflow in Figure 2, we used an error minimization procedure to estimate the refractive index of the grating material at the He-Ne laser wavelength, in which the numerical results of Equation (3) for $\eta_{0}^{*}\left(n\right)$, calculated for a measured AFM profile $h_{j}\left(x\right)$ and different discrete guesses of *n*, are compared with the average measured data $\eta_{0}^{exp}$. As a figure of merit, we used the adimensional parameter *ε*(*n*), defined as:(4)$$\varepsilon \left(n\right)=\left|\left(\frac{\eta_{0}^{*}\left(n\right)-\eta_{0}^{exp}}{\sigma_{\eta 0}}\right)\right|$$
where *σ_η_*_0_ is the standard deviation of the five independent measurements of *η*_0_. The best estimate for the refractive index is simply obtained as the value *n** that minimizes Equation (4), as shown graphically in Figure 3c. Finally, the best estimate $\bar{n}$ for the refractive index at the considered wavelength is the average of the *J* independent estimates of *n_j_* resulting from the different measured grating profiles $h_{j}\left(x\right)$. The standard deviation $\sigma_{n}=\sqrt{\sum_{j}{\left(\bar{n}-n_{j}^{*}\right)}^{2}/\left(J-1\right)}$, determined by the systematic profile variation due to grating inhomogeneities and defects, is used as the accuracy limit of the method (Figure 3d) and then as the (conservative) error limit for our refractive index estimates. At *λ* = 633 nm, we obtain with the proposed diffractive refractometer ${\bar{n}}_{633}=1.49\pm 0.01$, which is in full agreement with the accurate ellipsometry measurements of PMMA films $n_{633}^{ell}=1.491\pm 0.001$ [44]. It should be noted that a more homogeneous grating could have provided a higher accuracy for the estimation, allowing the use of an optimized grating depth to maximize sensitivity in the model and reduce systematic data scattering, so that the standard deviation of the mean $\sigma_{\bar{n}}=\sigma_{n}/\sqrt{J}\;$ of the *J* independent estimates could have been used as an error limit.

The accuracy of our estimate for ${\bar{n}}_{633}$ is additionally validated in Figure 3b by the close agreement of the measured diffraction efficiencies in the *m* ≠ 0 diffraction orders (blue bars) with the prediction of the scalar model Equation (3) (orange bars), calculated by using the best estimate ${\bar{n}}_{633}$ for the refractive index and an arbitrary measured surface profile *h_j_*(*x*) of the dataset.

To further maintain the strength and the versatility of the proposed diffractive refractometer, we used the same grating to measure the refractive index of PMMA at several discrete wavelengths *λ_p_* ranging from the visible to the NIR. For this purpose, we used a supercontinuum laser as a broadband light source and a set of different bandpass filters to sequentially select the narrow wavelength band (see Experimental Methods) for measuring the zero-order diffraction efficiency of the grating. In this configuration, schematized in Figure 4a, we repeated the fitting process using the same set of measured grating profiles $h_{j}\left(x\right)$ as in the previous analysis. Figure 4b shows the estimated refractive index *n*(*λ*) as compared with the ellipsometry data from the literature [44]. Again, the error in the estimate is dominated by the grating inhomogeneities rather than by the deviations introduced by the finite width of the spectral window in the model, showing a potential use of broadband lamps as light sources for the refractometer. Figure 4b shows the agreement with ellipsometry as highly evident in the entire spectral range, with a better match in the visible light due to the increased sensitivity of the method in this wavelength range for the actual grating used.

For a deeper insight into this aspect of the method performance, Figure 5 shows the combined parametric dependence of the sensitivity of the zero-order diffraction efficiency on the variations of the grating amplitude *h* ($S_{0,h}\left(\lambda ,h\right)=\left|\partial \eta_{0}\left(\lambda ,h\right)/\partial h\right|$) and light wavelength *λ* ($S_{0,\lambda }\left(\lambda ,h\right)=\left|\partial \eta_{0}\left(\lambda ,h\right)/\partial \lambda \right|$) in the scalar diffraction theory of a sinusoidal grating. To make such a dependence explicit from the dependence on *β* (Equation (2)) of the sensitivity $S_{0}=\left|\partial \eta_{0}\left(\beta \right)/\partial \beta \right|$ shown in Figure 1c, a model for the refractive index dispersion of the material is needed. According to the literature, [44] for the analysis in Figure 5, a Sellmeier model [9] for the refractive index n=n(λ) was used to describe the dispersion of the PMMA in the VIS-NIR spectral region, where it is transparent. Further details on this model, and the study of the effects of a more dispersive material (e.g., a TiO_2_ layer [50]) on the performance of the method also for higher diffraction orders, can be found in the Appendix A. The sensitivity dependence $S_{0,h}\left(\lambda ,h\right)$, shown in Figure 5a, allows the definition of the optimized grating amplitude *h* for maximized sensitivity at any target wavelength (as further illustrated by horizontal cross-sections at example wavelengths). Longer wavelengths of light generally require deeper surface relief gratings to operate at maximum efficiency. However, the method shows potentially better absolute accuracy in the visible compared to the NIR, even with grating amplitudes optimized for this regime. The wavelength sensitivity $S_{0,\lambda }\left(\lambda ,h\right)$, shown in Figure 5b, describes the effect of measuring the diffraction efficiency at different wavelengths with a single diffraction grating of fixed depth *h*. Its analysis is useful for characterizing the refractive index dispersion reconstructed with the diffractive refractometer using a single grating, as in the case of the experiment reported in Figure 4. From the analysis in Figure 5b, a general decreasing trend in sensitivity is observed from visible to near-infrared wavelengths is observed for PMMA grating modulation amplitudes below 0.6 μm, providing another basis to explain the NIR discrepancy in our experiment. If, on the one hand, the grating amplitude could be increased to work in this range with improved performance (e.g., as indicated by the brown horizontal cross section), on the other hand, wavelengths at zero sensitivity appear for higher grating modulations, preventing the broadband operation of the method.

In addition, the use of longer wavelengths with a single grating also leads to an increase in the λ/Λ ratio, which reduces the validity of the scalar approximation on which the proposed method is based, and further contributes to the reduced accuracy of the NIR data in the experiment of Figure 4.

These considerations suggest that the use of two or more diffraction gratings, with optimized grating amplitude and periodicity, could achieve better performance in retrieving the refractive index dispersion over a wider wavelength range with the proposed method. These two structural grating parameters are easily tunable in our direct writing configuration for master fabrication on azopolymer films, where the periodicity and the modulation depth of the grating are controlled via the irradiation dose of a tunable writing interferogram.

It is worth nothing, however, that even with the non-optimized broad-band performance of a single grating, the additional information about the refractive index dispersion of the material requires only trivial optical power measurements in our analysis.

## 4. Discussion

When evaluating the performance of the proposed diffractive method for the refractive index measurements of moldable solid thin films, one should consider the simplicity of the entire process, which involves very simple optical and topographical measurements without the need for multistep measurements (e.g., of the substrate alone) or modeling and dedicated expertise, as required for highly accurate measurements via ellipsometry.

Although the proposed diffractive spectrometer could not be the primary choice for the characterization of materials in general situations (anisotropic materials, multi-layers, non-transmissive materials, etc.), where ellipsometry still maintains a clear advantage, by dealing with the simultaneous and continuous characterization of both the refractive index and extinction coefficient of the material over a broader spectrum, the method has both theoretical and practical relevance. First, it is an elegant practical application of the simple scalar diffraction theory, here properly adapted here to describe the general complex light diffraction in a surface grating. Second, the method is effective for dielectrically isotropic and partially transmissive materials, which are often of practical interest in applications, such as in the fabrication of flat diffractive optical components. However, the method is also potentially applicable to monitor possible real-time variations of the grating material properties, provided that its topography is not altered. This capability arises from the use of the relative diffraction efficiency in relation to the refractive index extraction theory. Indeed, the dynamic evolution of the material index could be characterized by using an array detector (e.g., a CCD) to simultaneously record the power in all the propagating diffraction orders, and then by using their time-evolving relative diffraction efficiency. By correlating this information with the topography of the grating (e.g., previously measured), variations in the refractive index can be revealed. Furthermore, the use of an AFM is not mandatory for the method. Since the bottleneck for the accuracy of the method is mainly the homogeneity of the structural grating, recent holographic microscopes and optical profilometers [37], which are able to optically measure the topography of the surface with an accuracy of a few nanometers, could be used in spite of an AFM, achieving results similar to those presented in the manuscript.

## 5. Conclusions

In this work, we have proposed a theoretical and experimental framework for an operational diffraction-based method capable of measuring the refractive index of structured material layers from a simple optical and topographic characterization of a sinusoidal diffraction grating.

In addition to being applicable to structured surfaces where ellipsometry and other methods are ineffective, the proposed method has the advantage of being agnostic to several of the parameters relevant to conventional refractive index measurement of solid samples, including the film thickness, the substrate nature, and the surface roughness.

Since our approach is based on simple optical and topographical characterizations, it does not require dedicated equipment or specialized personnel and could be easily accessible to a broad audience working in optics, chemistry, and materials science, with reduced cost and high versatility.

## Figures and Tables

**Figure 1 polymers-15-01605-f001:**
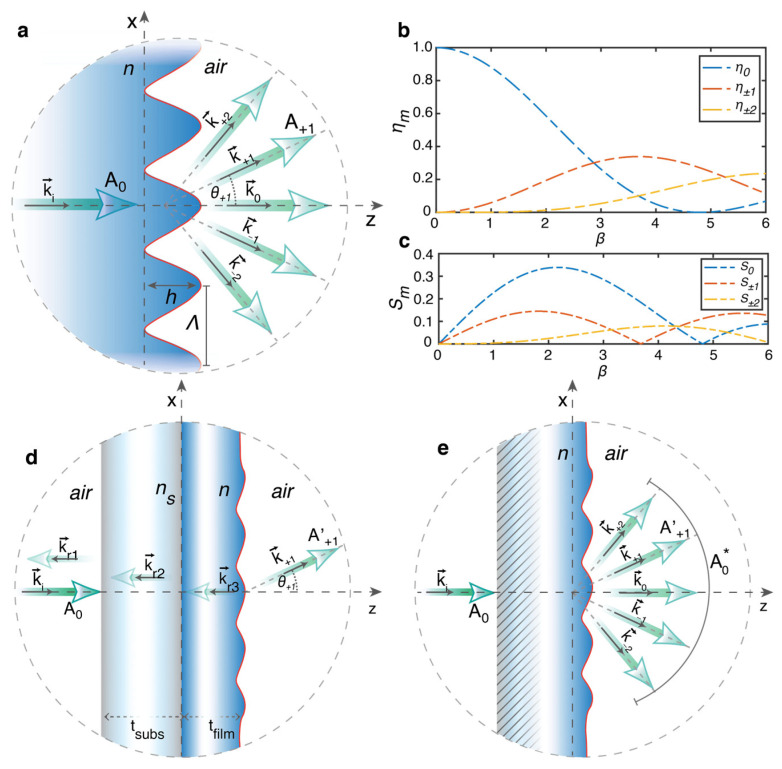
Diffraction behavior of an ideal sinusoidal surface relief grating within the scalar diffraction theory. (**a**) Scheme of the sinusoidal grating boundary at the interface of the infinite semi-spaces made of the dielectric material and air. (**b**) Scalar diffraction efficiency in the first five diffraction orders as a function of the maximum phase modulation parameter β. (**c**) Sensitivity of the diffraction efficiency variation in the first five orders as a function of β. (**d**) Simplified schematization of a real diffractive grating with finite thickness and multiple reflections (for simplicity, only the first backward reflected waves, with wavevectors k_ri_ are shown). Although the additional contribution to the transmitted diffracted field has low power (proportional to the product of the reflectance of the material-air and material-substrate interfaces), its inclusion prevents the analytical inversion of Equation (1). (**e**) Definition of the relative diffraction efficiency in the scalar diffraction model for a real grating configuration.

**Figure 2 polymers-15-01605-f002:**
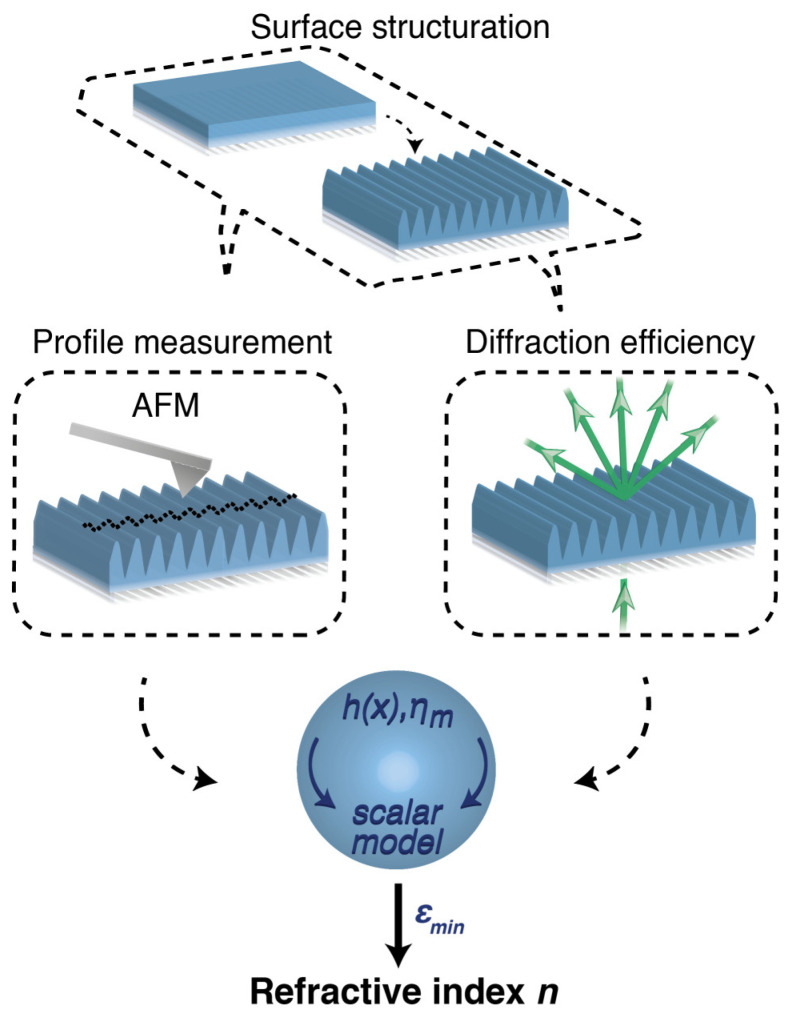
Workflow of the method. The diffractive refractometer based on the scalar theory involves structuring the material with a surface grating profile whose geometry satisfies the scalar approximation. The actual surface profile of the grating and the relative diffraction efficiency are measured and compared with the scalar diffraction model to estimate the best fit of *n*.

**Figure 3 polymers-15-01605-f003:**
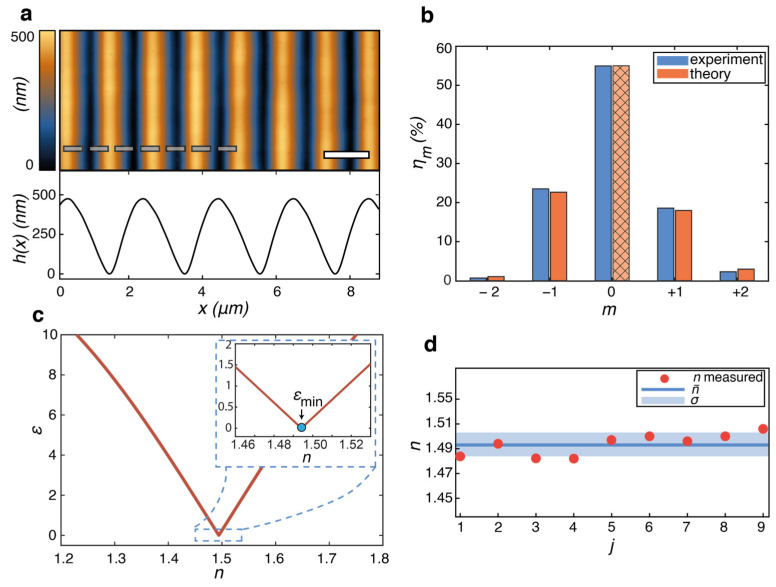
Measurement of the refractive index of a PMMA grating using the diffractive refractometer. (**a**) Typical AFM micrograph and surface profile *h*(*x*) in the measured grating region. (**b**) Measured (blue) and theoretical (orange) relative diffraction efficiencies in the first five orders. The theoretical efficiencies for *m* ≠ 0 are calculated using the refractive index estimate obtained from the fit of the zeroth order to the experiment, which requires minimization of the error parameter ε. (**c**) Typical behavior of the error *ε* parameter as a function of the *n* in the scalar model. The inset shows an enlarged view of the minimum region. (**d**) Average and standard deviation of the different *n* estimates resulting from the different measured grating profiles *h_j_*(*x*).

**Figure 4 polymers-15-01605-f004:**
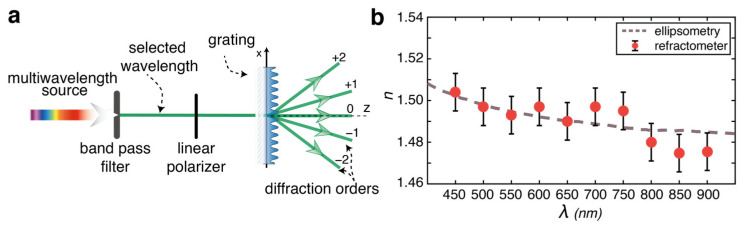
Measurement of the PMMA refractive index dispersion at discrete wavelengths using the diffractive refractometer. (**a**) Schematic of the optical setup. (**b**) Comparison of the measured refractive index *n*(*λ_p_*) and ellipsometry data from ref. [44].

**Figure 5 polymers-15-01605-f005:**
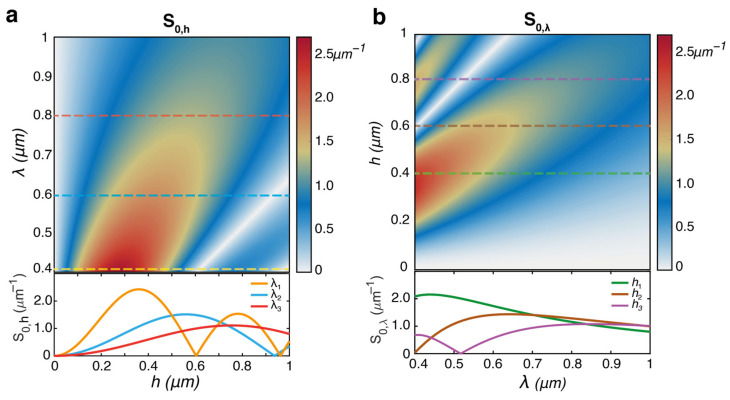
Simulated parametric diffraction sensitivity for PMMA in the 0-order of a sinusoidal surface relief grating. (**a**) Sensitivity $S_{0,h}\left(h,\lambda \right)$ of the efficiency variations to the parameter *h*. The plot shows the horizontal cross-sectional profiles (dashed lines) of the sensitivity $S_{0,h}\left(h\right)$ at *λ*_1_ = 0.4 μm (yellow), *λ*_2_ = 0.6 μm (blue), and *λ*_3_ = 0.8 μm (red). (**b**) Sensitivity $S_{0,\lambda }$ to the parameter *λ*. The cross-sectional profiles for $S_{0,\lambda }\left(\lambda \right)$ are calculated for *h*_1_ = 0.4 μm (green), *h*_2_ = 0.6 μm (brown), and *h*_3_ = 0.8 μm (violet).

## Data Availability

The data presented in this study are available upon request from the corresponding author.

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
