# Peer review of "Diffractive Refractometer Based on Scalar Theory"

_polymers, 2023, doi:10.3390/polym15071605_

Round 1

Reviewer 1 Report

See the attachment.

Author Response

Reviewer 1:

Salvatore et al. proposed a useful framework for an operating diffraction-based method able to measure the refractive index of structured material layers from simple optical and topographic characterization of a sinusoidal diffraction grating. This work is clearly presented and the conclusions are reasonable, the reviewer suggests that the work publish as it is or minor revision.

We thank the reviewer for the positive feedback on our work.

It would be better that the authors answer specific the following technical questions.

The authors chose only one polymer (PMMA) in the work, the reviewer is curious about the applicability of this method to other polymer films? Also, does the molecular weight of the polymer have an effect on the method?

According to the description included in the Introduction and Discussion section of the manuscript, the method we propose here is applicable to any semi-transparent material, provided that its surface is patterned with a diffraction grating with a geometry fitting within the boundaries of the scalar diffraction theory. For materials compatible with soft molding, the SRGs on azopolymer can be used as a master grating. However, any other surface structuring method can be used.

Material homogeneity and reduced birefringence are additional material requirements for applicability of the technique. The method is not directly affected by molecular weight.

The reviewer noticed that the relationship between n and ? is non-monotonic, could the authors make some explanations on this?

Refractive index dispersion (i.e. the dependence of n with λ) is a general characteristics of every material, depending on the electronic polarizability of the atoms. It is a generally non-monotonic function due to the presence of resonances at specific radiation frequencies.

The simplest model that explicitly describes the dependence for materials as composed by charged particles, is the Lorentz model (described in Ref 1 and many other optics-materials books). It also provides a general dependence of n(λ) that is compatible with the empirical Sellmeier equation used in the SI of our work.

There is some literature pertaining to the similar system that is of interest and echoes similar conclusions that the authors observed here: J. Phys. Chem. B 2015, 119, 10701−10709; J. Phys. Chem. B 2017, 121, 6416−6424.

We thank the reviewer for pointing out additional relevant literature. Accordingly, we have included the suggested references in the revised version of the manuscript.

Reviewer 2 Report

The authors show how the refractive index of a non-absorptive material can be directly measured, if its surface has a diffractive topography. As the catchy title already explains, their method is based on simple scalar diffraction theory.

The manuscript is excellent in its current form;  I recommend its publication as is.

I did find the article intriguing to read, because a) the proposed method is simple indeed, b) the authors not only clearly outline the limitations, but also c) critically assess/review their approach in detail. Furthermore, the methodology is supported by experimental data subjected to rigorous statistical analysis.

Author Response

The authors show how the refractive index of a non-absorptive material can be directly measured, if its surface has a diffractive topography. As the catchy title already explains, their method is based on simple scalar diffraction theory.

The manuscript is excellent in its current form; I recommend its publication as is.

I did find the article intriguing to read, because a) the proposed method is simple indeed, b) the authors not only clearly outline the limitations, but also c) critically assess/review their approach in detail. Furthermore, the methodology is supported by experimental data subjected to rigorous statistical analysis.

We thank the reviewer for the very positive feedback on every aspect of our work.